# Family Caregiver’s Loneliness and Related Health Factors: What Can Be Changed?

**DOI:** 10.3390/ijerph19127050

**Published:** 2022-06-09

**Authors:** Sylvie Bonin-Guillaume, Sylvie Arlotto, Alice Blin, Stéphanie Gentile

**Affiliations:** 1Neurosciences of Systems Institute, Institut National de la Santé et de la Recherche Médicale, UMR-Inserm 1106, Aix Marseille University, 13005 Marseille, France; sylvie.bonin@ap-hm.fr; 2Internal Medicine and Geriatric Department, Hôpitaux Universitaires de Marseille, Assistance Publique Hôpitaux de Marseille, 13005 Marseille, France; 3Service d’Evaluation Médicale, Assistance Publique Hôpitaux de Marseille, 13005 Marseille, France; alice.blin@ap-hm.fr (A.B.); stephanie.gentile@ap-hm.fr (S.G.); 4Health Service Research and Quality of Life Center (EA 3279), School of Medicine, La Timone Medical Campus, Aix Marseille University, 13005 Marseille, France

**Keywords:** loneliness, caregivers, frailty, caregiver burden

## Abstract

Background: Loneliness is a public health issue that may affect the entire population. Loneliness is associated with depression, sleep disorders, fatigue, and increased risk of obesity and diabetes. Risk factors for loneliness include having a poor social network and poor physical and mental health. The main objective was to study factors related to loneliness of family caregivers caring for independent older people. Methods: We performed a non-interventional observational cross-sectional study in south-eastern France. Family caregivers caring for people aged 70 and over and living at home were included. These older people were independent, without long-term conditions, and had applied for professional social assistance for daily living. Data were collected through a questionnaire, administered face-to-face or by telephone. Loneliness and perceived health status were measured through a single-question. Burden was assessed through the Mini-Zarit Scale, and frailty was measured through the Gerontopole Frailty Screening Tool. Results: Of the 876 family caregivers included, 10% felt lonely often or always. They reported more physical and mental health issues than those who did not feel loneliness (*p* < 0.001). Family caregivers with loneliness were more likely to be looking after a parent and were twice as likely to have a moderate to severe burden (OR = 2.6). They were more likely to feel anxious (OR = 5.6), to have sleep disorders (OR = 2.4), to be frail (OR = 2), and to view the status of their health as poor or bad (OR = 2). Conclusions: Loneliness has a negative impact on health, causes frailty, and places a burden on family caregivers. Means must be implemented to anticipate the consequences of the loneliness felt by family caregivers, notably by orienting them towards the relevant services.

## 1. Introduction

Loneliness is the subject of many investigations in public health because of its significant prevalence and impacts on health [1]. Loneliness is therefore a public health issue that may affect the entire population [2].

There are several different definitions of loneliness. The most commonly used is that of Perlman and Peplau [3], who describe loneliness as the gap between a person’s preferred and actual level of social contact, which is a subjective perception.

Researchers have distinguished loneliness from related concepts such as living alone and social isolation [1,4]. At its most basic level, social isolation has been defined as an objective state of having minimal social contact with other individuals (i.e., emotional loneliness) or to close friends and family (i.e., relational loneliness), which can be self-selected [5].

Perceived loneliness is measured using a variety of more or less standardized tools, leading to confusion about its prevalence in the literature [6].

Thus, the prevalence of loneliness among adults varies greatly because of the different tools used, the underlying concepts, and the populations investigated. In Europe, the average prevalence is 7%, but loneliness is unequally distributed across countries [7]. In France, the prevalence of loneliness is estimated in the general population as being between 10 and 19%, depending on the measurement tools used [8], compared to 27% of people aged 75 and over and 16% for caregivers of older people [9].

Loneliness is an independent risk factor for frailty [10]. Furthermore, loneliness is associated with depression, sleep disorders, and fatigue [11], with increased risk of obesity [12] and diabetes [13] and adverse outcomes such as functional decline and early death [14], emergency visits, or early admission in homecare [15]. Risk factors for loneliness include living alone [16], loss of a partner, poor social networks and resources, low socioeconomic status [17], and poor physical and mental health [18].

The factors associated with loneliness vary according to age and life stages [19]. Indeed, loneliness has a non-linear trajectory according to age and affects specific age groups, such as young adults or the oldest old persons, especially in relation to low income levels, greater functional limitations, or relational status.

Family caregivers (FCGs), especially those of older people, are particularly affected by loneliness [20]. FCGs are defined as people who provide unpaid, ongoing assistance to people with long-term conditions or limitations in terms of their activities of daily living (ADLs), such as grooming, feeding, bathing, walking, and dressing, or of instrumental activities of daily living (IADLs), such as shopping, meal preparation, housekeeping, and managing finances [14].

In France, there are 15 million FCGs [21], corresponding to 1 in 6 inhabitants.

Informal caregiving by FCGs is important to the health and social care systems because it supports those with a disability living in their own home for longer [22]. The reliance on such informal care will increase in the coming decades given the demographic projections of an aging world population [23].

Providing health and/or social care is not without consequences for FCGs, who can experience adverse impacts on their physical and mental health [24] and even decreased life expectancy [25], as a consequence. Factors associated with a higher burden on FCGs include the high number of hours spent caring, a poor relationship with the care recipient [26], or poor social networks [27].

Most studies on loneliness target specific populations such as older people or the vulnerable, e.g., those who are financially and socially vulnerable [28]. Few studies have focused on FCGs who have had their personal and social environment reduced due to the lack of time left after their professional life and their caregiving relationship [29]. It is therefore important to explore the loneliness of FCGs.

Thus, we hypothesized that the loneliness of FCGs is a factor related to their burden, health, and frailty. The main objective was to study factors related to the loneliness of family caregivers caring for independent older people.

## 2. Materials and Methods

### 2.1. Study Design

We performed a non-interventional observational cross-sectional study, in the Provence Alpes Côte d’Azur region (PACA), south-eastern France, between April 2016 and June 2017.

### 2.2. Study Population

We included FCGs caring for people aged 70 and over, living at home, independent according to the AGGIR grid (the Autonomie Gérontologie Groupes Iso-Ressources grid, which evaluates the independence of seniors and assigns a level between 1 and 6) [30], and without the long-term conditions and beneficiaries of the Caisse d’Assurance Retraite et de Santé Au Travail (Carsat). Carsat is a national pension and occupational health administration. To be included, the FCG had to have been designated by an older person who had applied to Carsat for social assistance for daily living.

### 2.3. Ethics Approval

This study was carried out in accordance with the bioethics laws, and the ethical approval of the Ethics Committee of the University of Aix-Marseille (No. 2020-09-07-10) was obtained. All personal data were managed exclusively by Carsat, which is authorized by the Commission Nationale de l’Informatique et des Libertés (CNIL) (No. 2005-38). In agreement with the Ethics Committee of the University of Aix-Marseille, a letter of information for FCGs was given during a Carsat visit to older people and/or sent to the FCG. A written receipt of consent was not required.

### 2.4. Variables and Measures

The data were collected from the FCG by a face-to-face or telephone interview. The detailed procedures were described in a previous publication but are summarized in brief here [27].

### 2.5. Judgement Criteria

The loneliness of FCGs was evaluated by a single-question (“Do you feel lonely?”) with a 5-point Likert scale (never, rarely, sometimes, often, or always) issued from the longer version of the center-epidemiological-study depression scale as it is a common and widely used measure of loneliness [5].

### 2.6. Other Variables Collected

All of the variables resulting from the questionnaire completed by the FCGs are not presented in this article but are presented in a previous publication [27].

Socio-demographic data of the FCGs were collected, such as age, gender, family situation, relationship with the care recipient (child, spouse, or other), professional activity, income, and family support in the daily life of the FCG.

In addition, variables related to FCGs’ health, frailty, and burden were collected.

To begin, the perceived health status of the FCGs was measured using a single 5-point Likert scale (from excellent to bad) and a 5-point Likert scale (from never to always), which were used to collect the existence of sleep disorders, anxiety, stress, or overwork. Finally, we asked FCGs if they had discussed their caregiving role with their physician. To complement the FCGs’ health status, we collected data about the presence or absence of long-term conditions, musculoskeletal disorders, falls in the past 6 months, regular physical activity, physician visits in the past 3 months, psychotropic drug use, and medical care foregone in the past year. In addition, the impact of the caregiving relationship on family life, outings, or even vacations was investigated.

FCGs’ frailty was assessed with 4 of the 6 items of the Gerontopole Frailty Screening Tool (GFST) because of the collection methods. Currently, no cut-off score has been determined [31].

FCGs’ burden was assessed with the Mini-Zarit scale [32,33]. The Mini-Zarit Scale consists of 7 items, scored on a 3-point Likert scale from 0 (never) to 1 (nearly always).

The total score ranges from 0 to 7 and therefore determines 4 levels of burden: absent or light (0-1), light to moderate (1.5-3), moderate to severe (3.5-5), or severe (5.5-7).

### 2.7. Statistical Analysis

All variables were examined through classical descriptive analysis. Categorical variables were described by their frequencies and percentages and ordinal or scale variables by their mean, standard deviation (±sd.), minimum, median, and maximum.

Dichotomizations were performed for three variables. The variable loneliness was thus recoded into “often/always” vs. “sometimes/rarely/never”. Frailty was recoded according to the median score obtained (lower or upper). Finally, burden was recoded into “moderate to severe/severe burden” versus the other two categories.

The recoded variable «loneliness» was described by univariate analysis. The associations between qualitative variables were measured by the Chi squared test and the exact Fischer test for small numbers. A Student Test or an Analysis of Variance (ANOVA) was performed for the ordinal or scales variables.

Multivariate logistic regression analysis was performed to test the independent significance of different variables. All variables with a threshold *p*-value of 0.1 were included in the model.

All statistical analyses were performed using SPSS (V.20.0, IBM, Armonk, NY, USA). The statistical tests were all two-sided, and statistical significance was indicated by a *p*-value of less than 0.05.

## 3. Results

Table 1 presents the sociodemographic characteristics of the FCGs and the comparison of these according to the presence or not of loneliness. The complete characteristics of the population have been published in a previous publication [27].

In our study, 10% of family caregivers felt lonely. FCGs who felt lonely lived significantly more lonely lives and identified a greater lack of family support.

Table 2 presents health, frailty, and burden profiles of the FCGs and the comparison of these according to their feeling of loneliness.

FCGs who felt lonely reported more physical and mental health issues, i.e., they were more likely to rate their health as poor or bad; to experience moderate to severe physical pain; and to feel anxious, stressed, or overworked. FCGs who felt lonely were more likely to be frail, according to the GFST scale; to experience moderate to severe burden; to have more of a negative impact on family life; and to have a bad relationship with older people.

The majority of the FCGs who felt lonely performed more than 3 tasks (80% of them versus 43%, *p* < 0.001) and had more difficulty performing them related to their own state of health, lack of financial or material means, lack of specialized institutions, or lack of dialogue with professionals (*p* < 0.05).

Multivariate analysis showed that family caregivers who felt lonely were more often children (OR = 1.8). In addition, they were twice as likely to have a moderate or severe burden (OR = 2.6) and were more limited in their assistance by lack of material or financial means (OR = 2.5). They were more likely to have a difficult relationship with older people (OR = 2.3) and more likely to have no family support (OR = 3.3).

They were 6 times more likely to feel anxious (OR = 5.6) and have sleep disorders (OR = 2.4), and twice as likely to be frail (GFST) (OR = 2) and experience their health as poor or mediocre (OR = 2) (Table 3).

## 4. Discussion

Nearly 10% of caregivers felt often or always lonely. Based on the literature review, this prevalence is similar to that found in the general population but slightly lower than that found among family caregivers [7,9].

However, the results confirm that loneliness has a negative impact on the health, frailty, and burden of FCGs. Indeed, analyses showed that FCGs who feel lonely are more likely to feel anxious, stressed, or overworked than others [34,35]. They are also more likely to perceive their health as poor or bad, and more likely to experience a moderate to severe burden despite being the same age as FCGs who do not feel lonely. Loneliness has also been associated with greater frailty, which is a known biopsychosocial risk factor for which early identification is advocated to prevent the physical and mental health consequences of FCGs.

This study highlights the need to collect the feelings of loneliness of FCGs. This could be done from a prevention perspective, upstream of the impacts on mental and physical health, burden, and frailty [34] and of proposed social prescriptions to avoid medical prescriptions, such as antidepressants and anxiolytics. Indeed, loneliness is a societal issue that can become a medical issue [1].

Additionally, it is important to understand factors predisposing one to feelings of loneliness in order to develop effective interventions to prevent and reduce it [36]. Indeed, preserving the good physical and mental health of FCGs is essential for the sustainability of the health care system [37] to limit the risk of unplanned hospitalization [32] or abuse of their care recipient [38].

In a prior study, the first action identified was to drive health professionals and social workers to identify who is a FCG and who is not, in order to be able to help them if needed. Identifying the FCGs and listening to them has been shown to be a first step in reducing their burden [27].

FCGs who feel lonely are more likely to be children who have a difficult relationship with their parent and limited family support. There is probably a certain precarious profile. Indeed, many articles reported the difficulties of working with FCGs, and FCGs are sandwiched between their professional life, their family life, and their role of FCG, which makes them more frail and is for some of them a source of social isolation and loneliness [39].

The health professional who identifies the issue of loneliness with the caregiver could thus be the initiator of a social prescription, in order to anticipate possible health issues. This social prescription could lead to actions such as participation in a shared garden, a support group, more professional help, support in administrative procedures, or even regular contact with the caregiver by a professional [40].

Numerous initiatives are in place to prevent loneliness and the social isolation of older people, but FCGs, who are at risk of stress and loneliness, are often excluded from these initiatives [41].

A possible limitation of this study is that FCGs included were caring for older people who asked for supplementary social assistance and professional help for their daily living. Yet, our results cannot be generalized to the entire elderly population assisted by FCGs because of this representative bias.

Loneliness is a phenomenon that is all the more topical as we have been living for a year and a half in a pandemic period, which is likely to persist for some time. This pandemic has accentuated the feeling of loneliness and the constraints that weigh on FCGs [42]. FCGs are therefore vulnerable in terms of physical and mental health and are particularly at risk when their needs are not addressed. The COVID-19 pandemic period resulted in increased pressure on FCGs and exacerbated their depressive symptomatology, especially among those who already felt lonely before the pandemic [43].

The COVID-19 pandemic also allowed for the development of human or virtual solutions, which can also evolve over time and be less expensive. However, it is necessary to include FCGs in these solutions and to develop actions specifically for them.

## 5. Conclusions

Loneliness is a societal issue that can become a medical issue. Physicians need to be informed and sensitive to prevent it, before having to manage medical consequences such as depression, frailty, stress, and the consumption of psychotropic drugs. Means must be implemented to anticipate the consequences of the loneliness felt by FCGs, notably by orienting them towards the relevant services.

## Figures and Tables

**Table 1 ijerph-19-07050-t001:** FCGs’ sociodemographic characteristics.

	Feeling Lonely (*Often or Always*) 10.3% (*n* = 91)	Not Feeling Lonely 89.6% (*n* = 785)	Total 876	*p*
Gender (female) * % (*n*)	73.6 (67)	63.4 (498)	64.5 (565)	0.055
Age (median ± SD)	61.4 ± 11.1	62.9 ± 13.9	62.7 ± 13.6	0.350
Living alone * % (*n*)	25.3 (23)	16.1 (126)	17 (149)	0.027
Child FCG * % (*n*)	67 (61)	59.9 (470)	60.6 (531)	
Spouse FCG % (*n*)	28.6 (26)	28.7 (225)	28.7 (251	0.106
Other FCG % (*n*)	4.4 (4)	11.5 (90)	10.7 (94)	
In professional activity % (*n*)	34.1 (31)	40.5 (318)	39.8 (349)	0.235
Low income ** % (*n*)	59.5 (50)	48.7 (342)	49.9 (392)	0.061
Helping for over a year % (*n*)	70.0 (42)	72.7 (381)	72.4 (423)	0.656
Without family support * % (*n*)	74.7 (68)	44.8 (352)	47.9 (420)	0.000

* variables included in the regression analysis. ** variable not included in the regression due to missing data.

**Table 2 ijerph-19-07050-t002:** FCGs’ health, frailty, and burden characteristics.

	FCGs Feeling Lonely (*Often or Always*) (*n* = 91)	FCGs Not Feeling Lonely (*n* = 785)	Total Population (*n* = 876)	*p*
**Health and frailty status**				
Having a chronic health problems * % (*n*)	52.7 (48)	39.6 (311)	41 (359)	0.016
Feeling his health status as poor or bad * % (*n*)	60.4 (55)	25.7 (202)	29.3 (257)	0.000
Feeling moderate/severe physical pain * % (*n*)	82.4 (75)	57.1 (448)	59.7 (523)	0.000
Complaining of sleep disorders * % (*n*)	73.6 (67)	28.2 (221)	32.9 (288)	0.000
Feeling anxious, stressed, or overworked% (*n*) *	80.2 (57)	19.9 (156)	26.1 (229)	0.000
Having a musculoskeletal disorder * % (*n*)	73.6 (67)	60 (471)	61.4 (538)	0.011
Being frail according to the GFST score * % (*n*)	67 (61)	28.9 (227)	32.9 (288)	0.000
Having no regular physical activity * % (*n*)	62.6 (57)	32 (251)	35.2 (308)	0.000
Having fallen during the previous 6 months * % (*n*)	22 (20)	13.5 (106)	14.4 (126)	0.029
Having consulted, at least once, a doctor for himself or herself during the previous 3 months * % (*n*)	89 (81)	79.4 (623)	80.4 (704)	0.028
Renouncing health care during the previous 12 months * % (*n*)	30.8 (28)	10.3 (81)	12.4 (109)	0.000
Taking at least one psychotropic drug * % (*n*)	46.2 (42)	21.1 (166)	23.7 (208)	0.000
Having talked with his doctor about his/her status as a FCG * % (*n*)	44 (40)	37.6 (295)	38.2 (335)	0.236
**Burden and perceived consequences of caregiving relationship**				
Having a moderate/severe burden (Mini-Zarit score) * % (*n*) *	78 (71)	30.6 (240)	35.5 (311)	0.000
Having a difficult relationship with older people * % (*n*)	42.9 (39)	13.2 (104)	16.3 (143)	0.000
Having an impact on family life * % (*n*)	54.9 (50)	22.9 (180)	26.3 (230)	0.000
Having an impact on the day’s outings * % (*n*)	69.2 (63)	43.2 (339)	45.9 (402)	0.000
Having an impact on leaving for a few days * % (*n*)	69.2 (63)	45.5 (357)	47.9 (420)	0.000
Perceiving difficulties in fulfilling their role as FCG * % (*n*)	83.5 (76)	49.9 (392)	53.4 (468)	0.000

* variables included in the regression analysis.

**Table 3 ijerph-19-07050-t003:** The factors associated with FCG loneliness.

	OR	95% CI		*p*
Child FCG *	1.846	1.048	3.250	0.034
FCG without family support *	3.379	1.837	6.123	0.000
FCG having a difficult relationship with older people *	2.328	1.300	4.168	0.004
Moderate/severe burden FCG *	2.604	1.387	4.890	0.003
FCG limited in assistance by the lack of material or financial means *	2.506	1.409	4.459	0.002
FCG complaining of sleep disorders *	2.476	1.355	4.524	0.003
FCG feeling anxious, stressed, or overworked *	5.634	2.987	10.629	0.000
FCG perceiving his health status as poor or bad *	2.457	1.282	4.708	0.007
Frailty according to the GFST score *	2.015	1.087	3.734	0.026

* A probability value of <0.05 was considered significant.

## Data Availability

The datasets used and/or analyzed during the current study are available from the corresponding author on reasonable request. These data will be used in other publications.

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
