# Peer review of "Family Caregiver’s Loneliness and Related Health Factors: What Can Be Changed?"

_ijerph, 2022, doi:10.3390/ijerph19127050_

Round 1

Reviewer 1 Report

I appreciate reviewing your valuable manuscript. I have some comments to improve it. Please reconsider and revise the following comments.

  1. You used a single question to measure the loneliness of FCGs which is the main variable. It was too simple to grasp the loneliness of the FCGs. Your reference did not show the appropriate evidence that the question was reliable to reflect your intent. Please reconsider and revise your manuscript.

  1. What would you like to explain in table 2? The total number of participants did not match your counts. Please reconsider and revise your manuscript.

  1. In table 2, you highlighted by the asterisk the variables which might be included in the multivariate analysis of table 3. However, the process of the multivariate regression analysis of your manuscript was unclear. So, the results in table 3 look inappropriate due to the lack of a clear explanation of the statistical analysis process. Please reconsider and revise your manuscript.

Author Response

Thank you very much for reviewing our manuscript.

Response point 1:

  • We are aware that using the single question to measure loneliness results in a subjective response. However, this choice was deliberate because loneliness is an individual subjective feeling. Furthermore, this single question has been used in many studies and Single-item self-report of loneliness has been shown to correlate highly with the two most widely used loneliness assessment tools, the UCLA Loneliness Scale (Russell, 1982) and the DeJong Gierveld Loneliness Scale (Van Baarsen, 2001).

Other references:

  • Kotwal AA, Cenzer IS, Waite LJ, Smith AK, Perissinotto CM, Hawkley LC. A single question assessment of loneliness in older adults during the COVID-19 pandemic: A nationally-representative study. J Am Geriatr Soc. 2022 May;70(5):1342-1345. doi: 10.1111/jgs.17700. Epub 2022 Feb 12. PMID: 35141875; PMCID: PMC9106870. https://pubmed.ncbi.nlm.nih.gov/35141875/
  • Patterson AC, Veenstra G. Loneliness and risk of mortality: a longitudinal investigation in Alameda County, California. Soc Sci Med. 2010 Jul;71(1):181-6. doi: 10.1016/j.socscimed.2010.03.024. Epub 2010 Mar 31. PMID: 20417589. https://pubmed.ncbi.nlm.nih.gov/20417589/
  • Savikko N, Routasalo P, Tilvis RS, Strandberg TE, Pitkälä KH. Predictors and subjective causes of loneliness in an aged population. Arch Gerontol Geriatr. 2005 Nov-Dec;41(3):223-33. doi: 10.1016/j.archger.2005.03.002. PMID: 15908025. https://pubmed.ncbi.nlm.nih.gov/15908025/
  • Theeke LA. Predictors of loneliness in U.S. adults over age sixty-five. Arch Psychiatr Nurs. 2009 Oct;23(5):387-96. doi: 10.1016/j.apnu.2008.11.002. Epub 2009 Jan 15. PMID: 19766930. https://pubmed.ncbi.nlm.nih.gov/19766930/
  • Tilvis RS, Pitkala KH, Jolkkonen J, Strandberg TE. Social networks and dementia. Lancet. 2000 Jul 1;356(9223):77-8. doi: 10.1016/s0140-6736(05)73414-0. PMID: 10892794. https://pubmed.ncbi.nlm.nih.gov/10892794/
  • Pinquart M, Sorensen S. Influences on loneliness in older adults: a meta-analysis. Basic Appl Soc Psychol. 2001;23(4):245–266. https://www.tandfonline.com/doi/abs/10.1207/S15324834BASP2304_2

Response point 2:

  • Table 2 described the differences among family caregivers who felt loneliness in terms of health, frailty status, burden, and perceived consequences of the caregiving relationship. Thus, we wished to investigate the impact of feeling lonely on the overall health profile of family caregivers. The total number of participants shown in Table 2 is 876, i.e. the entire cohort of our study. We have separated the results into 2 tables in order to simplify the understanding of our results.

Response point 3:

  • We indicated with an asterisk the variables that would be included in the multivariate analysis. We selected a threshold value of p-value at 0.1. The multivariate regression analysis process used in this manuscript was a binary logistic regression. Following your relevant comment, we have specified this detail in the "method" section of the manuscript. This clarification helps to understand the results shown in Table 3, which indeed describes the factors associated with FCGs loneliness.

We hope that these clarifications bring you the answers you were hoping for. Thank you again for reviewing our manuscript.

Reviewer 2 Report

I have reviewed the paper “Family caregiver’s loneliness and related health factors: what 1 can be changed?”

For the time of IJERPH, this paper is too descriptive. It does not provide novel results. The introduction is weak. The methodology corresponds to a study for journals of lower scientific impact.

The impact of the study on carers' loneliness is not clearly stated in the abstract and introduction.

The abstract does not contain the n of the actors.

I consider the subject to be of interest. However, I do not consider it a paper for the scientific rigour of IJERPH.

Author Response

We are aware that we are proposing the results of an observational study, however, we have focused on a population of family caregivers that is not yet well described in the literature. We were also able to confirm some of the results found in the literature for our specific population. This is the reason why we think that this manuscirt is valuable for the IJERPH.

It is true that we are not conducting a causal study here, but rather a description of the relationship between loneliness, burden, health and frailty of family caregivers. However, the causality between these different elements will be studied in a future publication on these family caregivers of the elderly, in particular on the longitudinal study of the impact of assistance in the daily life of the elderly on the burden, health, frailty and loneliness of their family caregivers.

We hope that these clarifications bring you the answers you were hoping for. Thank you again for reviewing our manuscript.

Reviewer 3 Report

This manuscript addresses the loneliness of caregivers of older adults, a topic that is often overlooked internationally. The study is well-designed with the use of easy-to-perform and commonly used measures. These measures are appropriate for this type of study and provide a broad picture of the factors that may contribute to loneliness in caregivers. Outcomes are not over-stated and the discussion is appropriate. 

The main question being asked, are there aspects of family caregiver loneliness that effect the health of caregivers as a population? A secondary question is whether those aspects are known to be modifiable. Because this is a non-interventional study, the secondary question is not pursued directly, but is addressed in the Discussion based on current knowledge. 

This topic is not new, in that caregiver stressors have been studies for decades. However, the authors address one key aspect that caregivers consistently acknowledge, loneliness, from a public health perspective. By focusing on loneliness as a key factor, the authors are able to determine the presence of loneliness in this population of elderly caregivers as well as the multiple factors that are related to being lonely. This research provides a basis for clinical research in multiple professions to address these factors to reduce loneliness with the goal of better care for onder adults.

While one could always argue for a more robust population, I believe that finding 876 people willing to participate in this study is very positive. The restrictions placed on enrollment are appropriate and reflect the larger population of family caregivers. While not all caregivers are elderly, older caregivers represent a more vulnerable population than do younger caregivers. I find the methodology appropriate for this older population. Given that the data were collected pre-COVID, one might find fault that all the interviews were not conducted in-person. I do not find the mix of in-person and telephone interviews to be problematic.

The conclusions do follow from the results. Data confirm that loneliness is present in about 10% of caregivers and that the health of these caregivers is negatively affected by the burden of caregiving loneliness. 

My only addition to my positive review is that I wish the authors had addressed the public health implications in more depth. Should these elderly caregivers receive additional supports when they are initially identified and should those supports span several years? Would home visits by a nurse, social worker, and/or physical therapy be used to prevent loneliness while providing additional guidance to support the caregiver? Are there policy implications whereby health insurance should cover the designated patient and the caregiver as a dyad rather than just covering the patient? Such a discussion moves beyond the data into the broader areas of public health.

Author Response

We thank you warmly for the precise and positive review of our manuscript. We also thank you for all the ideas you have suggested to us.

In particular, we will soon submit a manuscript on a longitudinal analysis of the impact of an elderly person's assistance plan on the burden, health and fragility of family caregivers. It is very relevant to underline that the caregiver has every interest in being included in this assistance plan, in the same way as the elderly person in the dyad that is being cared for.

Round 2

Reviewer 2 Report

I have seen again the revised paper that I had rejected, with my justification.
My decision remains the same. This article is not up to the standard of this Journal.

Author Response

Dear reviewer,
Thank you very much for the time spent on our manuscript.
Regarding your remarks:
You consider that the statistical analysis performed in our manuscript is purely descriptive, yet we performed a multivariate analysis and identified the factors associated with the feeling of loneliness (lines 164-170 and table n°3). 
Furthermore, you consider that the methodology of our study is not worthy of the journal, yet we followed the STROBE guidelines checklist (https://www.strobe-statement.org/checklists/) which guarantees the quality of the methodology and is required in high quality journals.
You consider the introduction to be weak, yet it contains 29 bibliographic references, explains the concepts of loneliness, and provides a literature review of the various studies conducted on the prevalence and associated factors of loneliness, 
It was this literature review that enabled us to identify that our study population, namely carers of independent older people, had not been the subject of any publications on this topic.   (line 76-79), .
We hope to have convinced you of the seriousness our work.

Kind regards